# Changes in Plasma VEGF and PEDF Levels in Patients with Central Serous Chorioretinopathy

**DOI:** 10.3390/medicina57101063

**Published:** 2021-10-05

**Authors:** Michał Chrząszcz, Weronika Pociej-Marciak, Katarzyna Żuber-Łaskawiec, Bożena Romanowska-Dixon, Marek Sanak, Katarzyna Michalska-Małecka, Mojca Globočnik Petrovič, Izabella Karska-Basta

**Affiliations:** 1Clinic of Ophthalmology and Ocular Oncology, Department of Ophthalmology, Faculty of Medicine, Jagiellonian University Medical College, 31-501 Kraków, Poland; m.a.chrzaszcz@gmail.com (M.C.); weronika.pociej@gmail.com (W.P.-M.); zuber.kasi@gmail.com (K.Ż.-Ł.); romanowskadixonbozena1@gmail.com (B.R.-D.); 2Molecular Biology and Clinical Genetics Unit, Department of Internal Medicine, Faculty of Medicine, Jagiellonian University Medical College, 31-501 Kraków, Poland; marek.sanak@uj.edu.pl; 3Department of Ophthalmology, Medical University of Silesia in Katowice, 40-055 Katowice, Poland; k.michalska.malecka@gmail.com; 4Eye Hospital University Medical Centre, Faculty of Medicine, University of Ljubljana, 1000 Ljubljana, Slovenia; mgpetrovic@yahoo.com

**Keywords:** angiogenesis, arteriogenesis, pachychoroid, central serous chorioretinopathy, VEGF, PEDF, CNV, AMD

## Abstract

*Background and Objectives:* Retinal pigment epitheliopathy and hyperpermeability of choroidal vessels were postulated to be involved in the pathogenesis of central serous chorioretinopathy (CSC). Imbalanced levels of vascular endothelial growth factor (VEGF) and pigment-epithelium–derived factor (PEDF) were previously implicated in the development of chorioretinal diseases characterized by increased vascular permeability. We aimed to compare the plasma levels of proangiogenic VEGF and antiangiogenic PEDF for 26 patients with acute CSC, 26 patients with chronic CSC, and 19 controls. *Materials and Methods:* VEGF and PEDF levels were measured using a multiplex immunoassay or enzyme-linked immunosorbent assay. Correlations with disease duration were assessed. *Results:* VEGF levels differed between groups (*p* = 0.001). They were lower in patients with acute CSC (*p* = 0.042) and chronic CSC (*p* = 0.018) than in controls. PEDF levels were similar in all groups. The VEGF-to-PEDF ratio was lower in CSC patients than in controls (*p* = 0.04). A negative correlation with disease duration was noted only for PEDF levels in the group with chronic CSC (rho = −0.46, *p* = 0.017). *Discussion:* Our study confirmed that patients with CSC have imbalanced levels of VEGF and PEDF. This finding may have important implications for the pathogenesis of CSC. VEGF-independent arteriogenesis rather than angiogenesis may underlie vascular abnormalities in these patients.

## 1. Introduction

Central serous chorioretinopathy (CSC) is a common early-onset chorioretinal disorder that affects mainly men between the ages of 20 and 60 years [1]. Accumulation of subretinal fluid results in visual disturbances including decreased visual acuity, central scotoma, metamorphopsia, and deterioration of vision-related quality of life [2].

Despite extensive research, the pathogenesis of CSC has not been fully clarified so far. Already in 1967, Gass [3] postulated that hyperpermeability of the choriocapillaris and increased hydrostatic pressure in the choroid contribute to retinal pigment epithelial (RPE) damage. This was later confirmed by studies employing indocyanine green angiography (ICGA) and deep range imaging optical coherence tomography, based on which CSC is currently classified as a pachychoroid disease [4]. Decompensation of RPE results in focal or diffuse breakdown of the outer retinal barrier, manifesting as retinal pigment epithelial detachment (PED), serous retinal detachment, and RPE atrophy [5,6]. RPE cells secrete different types of cytokines, including pigment-epithelium-derived factor (PEDF) and vascular endothelial growth factor (VEGF), which are involved in the development of various chorioretinal disorders, including CSC [7]. So far, the key role of these cytokines has been confirmed in the pathogenesis of age-related macular degeneration (AMD) [8,9,10,11]. Importantly, AMD and CSC share some similarities, such as disturbances in the levels of proangiogenic and antiangiogenic factors [12,13] and the presence of hyperpermeability. Moreover, they both can be complicated by choroidal neovascularization (CNV).

To the best of our knowledge, no previous studies have investigated changes in the circulating levels of PEDF in individuals with CSC. Therefore, we aimed to investigate whether CSC, as a pachychoroid disorder, may be associated with an imbalance of VEGF and PEDF levels. It is known that the balanced levels of antiangiogenic and proangiogenic cytokines are important for blood vessel function, while pachychoroid diseases are characterized not only by vascular abnormalities such as permeability but also by morphological vascular changes [6].

## 2. Materials and Methods

This case-control study included 52 adult white patients (11 female and 41 male) with acute CSC (*n* = 26) and chronic CSC (*n* = 26), diagnosed between November 2018 and June 2020 at the Department of Ophthalmology and Ocular Oncology of Jagiellonian University Medical College in Kraków, Poland. The control group consisted of 19 healthy volunteers recruited from the University Hospital in Kraków and matched for age, sex, hypertension, and smoking status with cases. CSC was diagnosed on the basis of characteristic features on indirect ophthalmoscopy, fluorescein angiography (FA), ICGA (SPECTRALIS, Heidelberg Engineering, Germany), and swept-source optical coherence tomography (SS-OCT) (DRI OCT Atlantis, Topcon, Japan). Patients with the following ocular conditions were excluded: uveitis, vasculitis, neovascular AMD, neovascular glaucoma, diabetic retinopathy, polypoidal choroidal vasculopathy, and other diseases causing macular exudation. Patients who underwent anti-VEGF therapy in the past as well as micropulse laser treatment, laser photocoagulation, or mineralocorticoid receptor antagonist therapy in the previous 6 months were also excluded. Systemic exclusion criteria were as follows: any malignancy, rheumatoid arthritis, psoriasis, any acute illness, kidney or liver dysfunction, or corticosteroid treatment, as well as stroke or acute myocardial infarction in the previous 6 months.

The study was approved by the Bioethical Committee of Jagiellonian University (opinion no. 122.6120.266.2016). All participants provided written informed consent to be included in the study.

### 2.1. Clinical Examination

All groups underwent best-corrected visual acuity measurement, indirect fundus ophthalmoscopy, and SS-OCT, while FA and ICGA were performed only in the patients with CSC. Acute CSC was defined as clinical signs and symptoms lasting less than 6 months, whereas chronic CSC was diagnosed when symptoms were present for 6 months or longer. On SS-OCT, the presence of PED and/or serous retinal detachment as well as increased central choroidal thickness (CT) was observed. On FA, a hyperfluorescent spot due to early-phase leakage that enlarged in the late phase (pooling) or hyperfluorescent granular widespread areas were present. Finally, on ICGA, areas of persistent hyperpermeability during the early and middle phases were observed, along with central hyperfluorescence during the late phase.

### 2.2. Sample Collection

Morning blood samples were obtained from all 71 participants (in women of fertile age only in the proliferative phase of the menstrual cycle). Samples were collected from the antecubital vein into BD Vacutainer (BD Life Sciences, Franklin Lakes, NJ, USA). EDTA tubes were used for plasma preparation. Plasma VEGF levels were measured using the Human Angiogenesis A Premixed Mag Luminex Performance Assay (FCSTM02-10, R&D Systems, Minneapolis, MN, USA), which contains premixed fluorogenic beads with monoclonal antibodies against VEGF. The measurements were performed in line with the manufacturer’s protocol with a plasma dilution of 1:4 and the xMAP analyzer (Luminex Corporation, Austin, TX, USA). A bead-trapped cytokine was detected by biotin-streptavidin sandwich immunocomplex fluorescence. The results were calculated using 7-point standard curves and proprietary software, Milliplex Analyst Version 5.1 (Merck, Darmstadt, Germany). Finally, PEDF levels were measured at a plasma dilution of 1:2 using PEDF ELISA Kit (Biomatik, Wilmington, DE, USA) according to the manufacturer’s protocol.

### 2.3. Statistical Analysis

For qualitative data, numbers and percentages were used. Quantitative data were presented as means and standard deviations (SDs) for normally distributed variables and as medians and interquartile ranges (IQRs) for variables without normal distribution. For the distribution of quantitative variables, the Kolmogorov–Smirnov test was used. Qualitative variables were compared between groups using the Pearson χ^2^ test. The test was used when expected frequencies in more than 80% of cells were higher than 5. The Fisher–Freeman–Halton test was used otherwise. Quantitative variables were compared between groups using the one-way analysis of variance for normally distributed variables and the Kruskal–Wallis test for variables without normal distribution. If the comparison between the three groups yielded a significant *p* value, a pairwise comparison with the Bonferroni correction was applied. The comparison between the whole CSC group (both acute and chronic) and controls was made using the t test or Mann–Whitney test, as appropriate. A box plot was applied to graphically present the results, with the line inside the box representing a median; the lower and upper sides of the box representing the lower and upper quartiles, respectively; the horizontal lines connected to the box with vertical lines representing cases distant up to 1.5 of the IQR from the respective quartiles; circles representing cases distant from 1.5 to 3 IQRs from the respective quartile; and asterisks representing cases distant by more than 3 IQRs from the respective quartile. Finally, the Spearman rank correlation coefficient was used to assess the strength of the relationship between quantitative variables, while a scatterplot with the locally weighted smoothing curve was used to assess the shape of the relationship. A *p* value of less than 0.05 was considered significant. IBM SPSS Statistics 26 for Windows was used for the statistical analysis.

## 3. Results

### Characteristics of Patients

In the group with acute CSC, men constituted 80.8%, as compared with 76.9% in the group with chronic CSC and 57.9% in the control group. The mean (SD) age of patients with acute CSC was 43.5 (9.8) years; of patients with chronic CSC, 44.0 (5.8) years; and of controls, 39.1 (7.5) years. Table 1 summarizes the demographic and clinical characteristics of the groups. There were no differences between patients with CSC and controls in terms of sex, age, the prevalence of systemic hypertension, and smoking status.

The following medications were used in the study groups: angiotensin-converting enzyme inhibitors (three patients with acute CSC and three controls); β-blockers (one patient with chronic CSC and one control); calcium channel blockers (four patients with acute CSC and one patient with chronic CSC); diuretics (five patients with acute CSC, four patients with chronic CSC, and one control); and sartans (one patient with acute CSC and two patients with chronic CSC). No significant differences in PEDF and VEGF levels were observed between patients using at least one antihypertensive drug and those not receiving any antihypertensive medication (data not shown). Among the eleven women with CSC, one received hormone replacement therapy and two were after menopause.

The ophthalmological characteristics of the study groups are presented in Table 2. There were significant differences between groups in terms of best-corrected visual acuity (Table 2).

Plasma VEGF levels differed between all study groups (*p* = 0.01) (Figure 1). Pairwise comparisons showed that plasma VEGF levels were lower in patients with acute CSC (*n* = 26) than in controls (*n* = 19) (*p* = 0.04). Similarly, VEGF levels were lower in patients with chronic CSC (*n* = 26) than in controls (*p* = 0.02). Additionally, the analysis of all the CSC patients (both acute and chronic form (*n* = 52)) in comparison with controls (*n* = 19) showed stronger significance (*p* = 0.00) (Figure 2). PEDF levels did not differ between the three groups (*p* = 0.50) or between CSC patients (*n* = 52) and controls (*n* = 19) (*p* = 0.83). The plasma levels of VEGF and PEDF in study groups are shown in Table 3.

The above results were confirmed by a significantly downregulated VEGF-to-PEDF ratio in patients with CSC compared with controls (Figure 3).

Importantly, a negative correlation between disease duration in patients with chronic CSC and PEDF levels was noted (rho = −0.46, *p* = 0.017) (Figure 4), while no such correlation was observed for VEGF levels.

## 4. Discussion

RPE cells produce a range of angiogenic factors, including VEGF and PEDF, which are strongly related to the development of chorioretinal diseases [7]. VEGF is a key factor promoting angiogenesis and vascular permeability [13]. It is a mitogen and chemokine of vascular endothelial cells responsible for cell division and proliferation. VEGF and PEDF act as functional antagonists [14]. PEDF belongs to the serine protease inhibitor family and exerts various biological functions through antioxidant, antiapoptotic, and anti-inflammatory activity. It also inhibits angiogenesis and elevated vascular permeability [15,16].

To our knowledge, we are the first to describe the plasma levels of VEGF and PEDF—two counteracting cytokines—in patients with CSC in comparison with healthy controls. In line with our previous research [17], in the current study, VEGF was significantly lower in patients with acute and chronic CSC than in the control group. Plasma VEGF levels differed between patients with acute CSC, those with chronic CSC, and healthy individuals. The difference was more significant when the whole CSC group was compared with controls.

Some previous studies described the involvement of plasma VEGF in the pathophysiology of CSC [12,17,18]. Lim et al. [18] described similar plasma VEGF levels in patients with CSC and controls, but the study included a small sample size. Terao et al. [19] reported that VEGF levels were downregulated at levels that almost reached significance, but only along with CSC progression from the acute to chronic form.

Our results may support the hypothesis that vascular abnormalities in CSC are related more to arteriogenesis than hypoxia-independent angiogenesis. As previously reported by Spaide et al. [20] and Sacconi et al. [21], CNV in patients with CSC is caused by proliferation of new vessels during arteriogenesis. The characteristic feature of this process is dilation of the existing vascular channels. Moreover, the process is not dependent on VEGF in contrast to highly VEGF-dependent angiogenesis [22,23].

Based on our results showing downregulated VEGF levels in patients with CSC, we speculate that the triggering mechanisms in CSC differ from those in AMD, despite similar clinical manifestations (as best revealed by ICGA [24]). Patients with CSC are generally younger than those with AMD; therefore, different mechanisms may underlie the pathogenesis of both entities. In AMD, hypoxia leads to elevated autophagy in RPE cells, and expression of VEGF and PEDF might be regulated by autophagy on exposure to hypoxia to further participate in the formation of retinal neovascularization [7,25]. The downregulated levels of VEGF in CSC patients vs. healthy controls described in the current study may partially explain the unsatisfactory efficacy of intravitreal anti-VEGF treatment in patients with CSC [26,27,28]. Although the available data are discordant, a number of previous studies showed the superiority of low-fluence photodynamic therapy over anti-VEGF treatment in patients with CSC [29,30].

In our study, we did not observe differences in plasma PEDF levels between patients with CSC and controls. On the other hand, it was shown that abnormal PEDF levels may play an essential role in the modulation of vascular leakage in some other maculopathies [31,32]. Wang et al. [32] described impaired production of trombospondin-1 and PEDF in diabetic macular edema. Machalińska et al. [8] demonstrated decreased PEDF levels in dry AMD and a strong positive correlation between plasma VEGF and PEDF levels in neovascular AMD. Our results are in line with those reported by Chen et al. [33], who measured PEDF levels in patients with high myopia, which is, similarly to CSC, associated with RPE dysfunction. They reported no differences in serum PEDF levels between patients with myopia and controls. However, differences were reported for aqueous humor PEDF levels. This may suggest that aqueous PEDF levels in CSC, as in high myopia, are not determined by serum PEDF levels but rather by alterations in intraocular PEDF synthesis. This indicates the need for further research on the balance between proangiogenic and angiogenic factors in intraocular fluids and their role in the pathogenesis of CSC.

The PEDF 44-mer peptide was reported to counteract the VEGF-induced increase in vascular permeability [14]. Interestingly, in our study, the VEGF-to-PEDF ratio was lower in all CSC patients than in controls, which is a result of downregulated VEGF levels and stable PEDF levels. This may suggest impaired balance between plasma proangiogenic and antiangiogenic factors in the course of CSC.

Of note, we found a negative correlation between plasma PEDF levels and disease duration, which may point to a link between degenerated RPE and impaired production of PEDF in long-term chronic CSC [33]. The presence of widespread diffuse areas of RPE atrophy on fundus autofluorescence was reported in chronic CSC [34], and the chronic form of CSC had been initially termed “diffuse pigment epitheliopathy” [6]. Furthermore, in many cases of advanced chronic CSC (lasting longer than 5 years), deep range imaging optical coherence tomography revealed intraretinal cystoid edema, possibly indicating the lack of PEDF activity [35].

Our research was performed at a single time point and, in addition, only plasma samples were investigated, which constitutes a limitation of the study.

Importantly, VEGF and PEDF perform numerous functions in the body that are responsible for maintaining systemic homeostasis [36,37]. Therefore, the plasma levels of angiogenic factors may not reflect their levels in the chorioretinal complex. Further studies, including those investigating levels of VEGF and PEDF in the vitreous body, are needed to fully elucidate the role of angiogenic factors in the pathogenesis of CSC.

## 5. Conclusions

The results of our study suggest that downregulated plasma levels of VEGF play a more important role in the pathogenesis of CSC than PEDF, the levels of which remain stable throughout the disease course. Overall, this imbalance of otherwise normally counteracting cytokines may have significant implications for future research on CSC. For now, the pathomechanisms underlying this condition remain largely unknown, but it can be speculated that intraocular lesions may be related more to alterations in intraocular cytokine levels than to systemic abnormalities. More research is needed to further elucidate these complex issues.

## Figures and Tables

**Figure 1 medicina-57-01063-f001:**
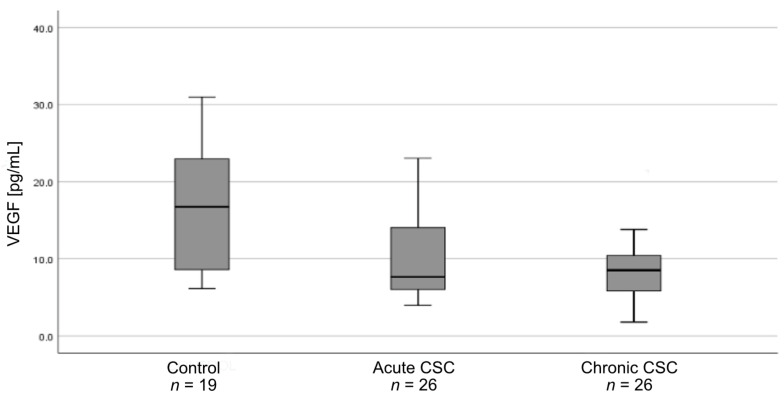
Box-and-whisker plot of plasma vascular endothelial growth factor (VEGF) levels in patients with acute central serous chorioretinopathy (CSC), patients with chronic CSC, and controls (*p* = 0.01).

**Figure 2 medicina-57-01063-f002:**
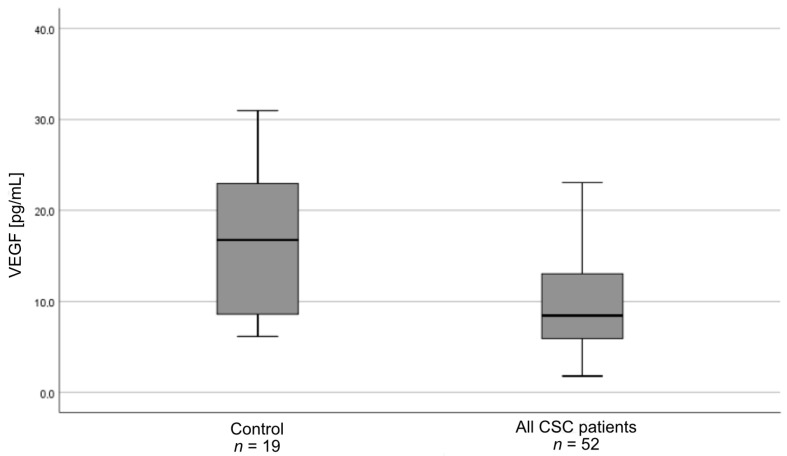
Box-and-whisker plot of plasma vascular endothelial growth factor (VEGF) levels in all patients with central serous chorioretinopathy (CSC) (acute and chronic) as well as controls (*p* = 0.003).

**Figure 3 medicina-57-01063-f003:**
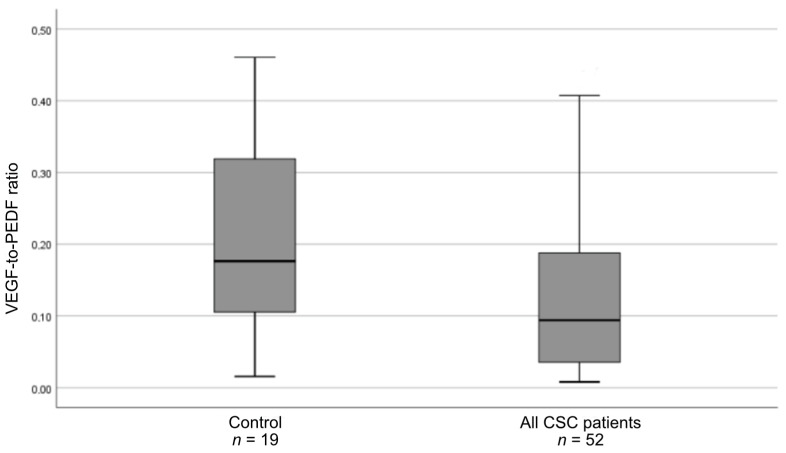
Box-and-whisker plot of the ratio of VEGF to pigment-epithelium–derived factor (PEDF) in all patients with central serous chorioretinopathy and controls (*p* = 0.04).

**Figure 4 medicina-57-01063-f004:**
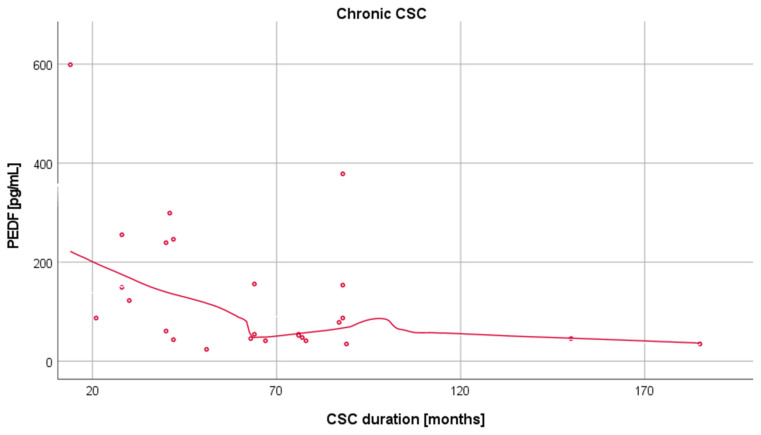
Correlation between the plasma levels of PEDF and disease duration in patients with chronic CSC.

**Table 1 medicina-57-01063-t001:** Demographic and clinical characteristics of patients with acute central serous chorioretinopathy (CSC), patients with chronic CSC, and controls.

Parameter	Chronic CSC (*n* = 26)	Acute CSC (*n* = 26)	Controls (*n* = 19)	*p* Value
Male sex, *n* (%)	20.0 (76.9)	21 (80.8)	11.0 (57.9)	0.20
Age, y	44.0 (5.8)	43.5 (9.8)	39.1 (7.5)	0.09
Current or former smoking, *n* (%)	7.0 (26.9)	6 (23.1)	5.0 (26.3)	0.94
Hypertension, *n* (%)				
Allergy	4.0 (15.4)	10.0 (38.5)	4.0 (21.1)	0.14
Psychiatric disorders	2 (7.7)	2 (7.7)	1 (5.3)	0.93
Hashimoto disease	1 (3.8)	1 (3.8)	0	0.92
Gout	2 (7.7)	1 (3.8)	0	0.81
*Helicobacter pylori* infection	2 (7.7)	1 (3.8)	0	0.81
	5 (19.2)	1 (3.8)	0	0.17

Data are expressed as number (percentage) unless otherwise specified. A *p* value <0.05 was considered significant.

**Table 2 medicina-57-01063-t002:** Ophthalmological characteristics of patients with acute central serous chorioretinopathy (CSC), patients with chronic CSC, and controls.

Parameter		Chronic CSC (*n* = 26)	Acute CSC (*n* = 26)	Controls (*n* = 19)	*p* Value
Affected eye, *n* (%)	Right	9 (34.6)	13 (50.0)	-	0.412
	Left	11 (42.3)	10 (38.5)	-	
	Both	6 (23.1)	3 (11.5)	-	
BCVA, *n* (%)	0.5 < * ≤ 1.0	21 (80.8)	17 (65.4)	19 (100.0)	0.016
	0.1 ≤ * ≤ 0.5	5 (19.2)	9 (34.6)	0	

Data are expressed as number (percentage) unless otherwise specified. A *p* value <0.05 was considered significant. Abbreviations: BCVA, best corrected visual acuity, * BCVA result.

**Table 3 medicina-57-01063-t003:** Plasma levels of vascular endothelial growth factor (VEGF) and pigment-epithelium-derived factor (PEDF) in patients with acute central serous chorioretinopathy (CSC), patients with chronic CSC, and controls.

Angiogenic Factor, pg/mL	Acute CSC (*n* = 26)	Chronic CSC (*n* = 26)	Controls (*n* = 19)	*p* Value
VEGF	7.70 (6.00–14.40)	8.50 (5.80–10.40)	16.80 (8.50–23.00)	0.01
PEDF	94.99 (24.45–278.43)	69.75 (45.73–155.99)	76.31 (56.23–167.18)	0.50

Data are expressed as median (interquartile range). A *p* value <0.05 was considered significant. Abbreviations: VEGF, vascular endothelial growth factor; PEDF, pigment-epithelium–derived factor.

## Data Availability

Not applicable.

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
