# Peer review of "Changes in Plasma VEGF and PEDF Levels in Patients with Central Serous Chorioretinopathy"

_medicina, 2021, doi:10.3390/medicina57101063_

Round 1
Reviewer 1 Report
The manuscript presents the original clinical data and is of interest for delineating the mechanism of retinal disease. Please cite some of the work on PEDF from Dr. Tombran-Tink, who co-discovered PEDF and did a lot of work on PEDF in retina and RPE in particular. Please have the manuscript edited by somebody who is an authentic English language speaker.
Author Response
Thank you for your insightful comments. We have now revised our manuscript according to your suggestions. We hope that you approve of the current version.
We added the reference. Ref .[16]
Tombran-Tink J. PEDF in angiogenic eye diseases. Mol. Med. 2010, 10: 267–278. doi: 10.2174/156652410791065336.
Page:7; line: 206.
Reviewer 2 Report
The paper investigates the plasma levels of VEGF and PEGF in patients with central serous chorioretinopathy vs normal subjects. The subject is original and of interest, and it may be a starting point for new therapeutic choices, as by example the use of intravitreal anti-VEGF. The paper is well documented.
However, In the body, VEGF is stimulated by ischemia, inflammation or trauma, and has a local, tissue, autoid action in regulating the phenomena of angiogenesis and wound repair and maintaining organ health. It is quickly captured by platelets and inactivated in the general circulation. The clinical significance of free circulating VEGF level is challenging to establish due to its local, tissular mechanism of action
As minor observation, I would recommend:
1. please add in the discussion section, as a limitation of the study, the fact that the plasma levels of VEGF may not reflect the local retinal and choroidal levels. Dosing the VEGF in the vitreous body could provide a better approach
Author Response
Thank you for your insightful comments. We have now revised our manuscript according to your suggestions. We hope that you approve of the current version.
Your comprehensive analysis allowed us to address the obvious limitations of our study.
Page:8; line: 266-271.
Importantly, VEGF and PEDF play numerous functions in the body that are responsible for maintaining systemic homeostasis [36,37]. Therefore, the plasma levels of angiogenic factors may not reflect their levels in the chorioretinal complex. Further studies, including those investigating levels of VEGF and PEDF in the vitreous body, are needed to fully elucidate the role of angiogenic factors in the pathogenesis of CSC.
This manuscript is a resubmission of an earlier submission. The following is a list of the peer review reports and author responses from that submission.